# On the Potential of Improving WRF Model Forecasts by Assimilation of High-Resolution GPS-Derived Water-Vapor Maps Augmented with METEOSAT-11 Data

**Anton Leontiev [1], Dorita Rostkier-Edelstein [2,3] and Yuval Reuveni [4,5,6,*]**

[1]  Department of Electrical Engineering, Ariel University, Ariel 40700, Israel; antonle@ariel.ac.il
[2]  Department of Environmental Physics, IIBR, Ness-Zyiona 74100, Israel;
     dorita.rostkier-edelstein@mail.huji.ac.il
[3]  The Fredy and Nadine Herrmann Institute of Earth Sciences, The Hebrew University of Jerusalem,
     Rehovot 7610001, Israel
[4]  Department of Physics, Ariel University, Ariel 40700, Israel
[5]  Eastern R&D Center, Ariel 40700, Israel
[6]  School of Sustainability, Interdisciplinary Center (IDC) Herzliya, Herzliya 4610101, Israel
*  Correspondence: yuvalr@ariel.ac.il; Tel.: +972-74-7296725 or +972-52-5970648; Fax: +972-3-9366834

**Abstract:** Improving the accuracy of numerical weather predictions remains a challenging task. The absence of sufficiently detailed temporal and spatial real-time in-situ measurements poses a critical gap regarding the proper representation of atmospheric moisture fields, such as water vapor distribution, which are highly imperative for improving weather predictions accuracy. The estimated amount of the total vertically integrated water vapor (IWV), which can be derived from the attenuation of global positioning systems (GPS) signals, can support various atmospheric models at global, regional, and local scales. Currently, several existing atmospheric numerical models can estimate the IWV amount. However, they do not provide accurate results compared with in-situ measurements such as radiosondes. Here, we present a new strategy for assimilating 2D IWV regional maps estimations, derived from combined GPS and METEOSAT satellite imagery data, to improve Weather Research and Forecast (WRF) model predictions accuracy in Israel and surrounding areas. As opposed to previous studies, which used point measurements of IWV in the assimilation procedure, in the current study, we assimilate quasi-continuous 2D GPS IWV maps, combined with METEOSAT-11 data. Using the suggested methodology, our results indicate an improvement of more than 30% in the root mean square error (RMSE) of WRF forecasts after assimilation relative standalone WRF, when both are compared to the radiosonde measured data near the Mediterranean coast. Moreover, significant improvements along the Jordan Rift Valley and Dead Sea Valley areas are obtained when compared to 2D IWV regional maps estimations. Improvements in these areas suggest the impact of the assimilated high resolution IWV maps, with initialization times which coincide with the Mediterranean Sea Breeze propagation from the coastline to highland stations, as the distance to the Mediterranean Sea shore, along with other features, dictates its arrival times.

**Keywords:** integrated water vapor; GPS; METEOSAT; weather research and forecast; data assimilation

## 1. Introduction

One of the most wide-spread natural greenhouse gases, which constantly exists in the atmosphere is water vapor (WV) [1]. Integrated water vapor (IWV) is defined as the vertically integrated amount of WV and can be expressed in kg/m$^2$, or as precipitable water (PW), defined by the height of an equivalent liquid water column in millimeters [2,3]. Since a significant number of hydrological response features to warming are a direct result of the increase in lower-tropospheric WV amounts, acquiring WV distribution, both in space and time, is essential for studying the hydrological cycle. Therefore, it can be used as a governing factor in climatological studies at different spatial scales). In addition, IWV can

be used for estimating humidity at different height levels, and consequently also augment hydrological and climate modeling at regional and local scales as it is commonly used in evapotranspiration estimations and energy balance assessments [4]. However, due to the spatio-temporal moisture field high variability, it is poorly represented and considered as a less well described parameter in the initial conditions of Numerical Weather Prediction (NWP) models [5], such as the Weather Research and Forecast model (WRF) [6,7].

There are several techniques for assessing WV amount in the troposphere. The most common one is using radiosondes via in situ measurements [8–10]. Radiosondes provide measurements of pressure, temperature, and relative humidity, which enable deriving water vapor mixing ratio and other related atmospheric parameters as a function of altitude [11].

During the beginning of the 1990s, geophysicists and geodesists have made it possible to retrieve the amount of WV in the troposphere, by developing methods for measuring the degree to which signals, emitted from global navigation satellite systems (GNSS) and propagates to global positioning systems (GPS) base station receivers, are delayed by atmospheric WV molecules [12]. This delay is parameterized in terms of a time-varying zenith wet delay (ZWD) that is retrieved by stochastic filtering of the GPS raw measurements [2,3,13]. The ZWD data possess vertically integrated information regarding the atmospheric refractivity index which is a function of the WV pressure, temperature and atmospheric pressure [14]. Today, GNSS meteorology can deliver continuous estimation of WV amount in the troposphere, with high spatio-temporal resolution, assuming the temperature and pressure values could also be recovered at the observation sites. Numerous GNSS-based estimates of WV have been compared with radiosondes data [15], WV radiometers (WVR) measurements [16] or ECMWF meteorological analysis fields [17]. The results indicate that WV estimations retrieved by GNSS are generally in good agreement with measurements from radiosondes and WVR [18–20].

For the last two decades, applications of IWV estimations derived from GNSS tropospheric path delays have focused on several main directions such as: validations of IWV measurements by radiosondes or remote sensing satellite sensor sets, such as GOME-2, MODIS, OMI, SEVIRI and AIRS, combined with IWV estimations by NWP models [21,22]; climate change studies and weather forecasting [23]; assimilation of GNSS-IWV estimations in NWP models, such as WRF model [24–27]. These studies used point measurements data assimilation, from radiosondes measurements or GPS ZWD estimations, leading to small improvements (5–10%), for example [24,27] in WRF forecasts.

Here, we investigate the potential of improving WRF model forecasts by assimilation of high-resolution 2D IWV distribution maps derived from GPS tropospheric path delays augmented by METEOSAT-11 WV imagery data, based on the survey of Israel–active permanent network (SOI-APN) with more than 20 GPS geodetic stations over the entire country (Figure 1).

Estimation of WV amount in the troposphere using remote sensing measurements from satellites such as the METEOSAT series is useful for producing proper 2D IWV distribution maps [28–30]. Recently, Leontiev and Reuveni [31,32] developed a technique for augmenting IWV estimations using both remote sensing measurements from satellites such as METEOSAT and GNSS tropospheric path delays. The suggested strategy is based first on estimating METEOSAT 7.3 μm WV pixel values by extracting the mathematical dependency between the IWV amount extracted from GPS ZWD and the METEOSAT-10 data. Since the METEOSAT-10 data was available with the METEOSAT-11 for 2018 year, the METEOSAT-11 data was used in the current research. The surface temperature differences between ground station measurements and METEOSAT 10.8 μm infra-red (IR) channel is then used to identify spatio-temporal cloud distribution structures. The classified cloud structures are then projected into the GPS-IWV estimation map while interpolating between adjacent GPS station inside the network. This method increases the accuracy of the IWV regional map estimations being verified against in situ radiosonde measurements, and assists to acquire the absolute amount of water in the atmosphere, both in the form of clouds and vapor. The application of this method reduced the mean error (ME) and root

mean square (RMSE) differences between the GPS-IWV estimations and the radiosonde measurements from 1.77 and 2.81 kg/m² to 0.74 and 2.04 kg/m², respectively [31,32].

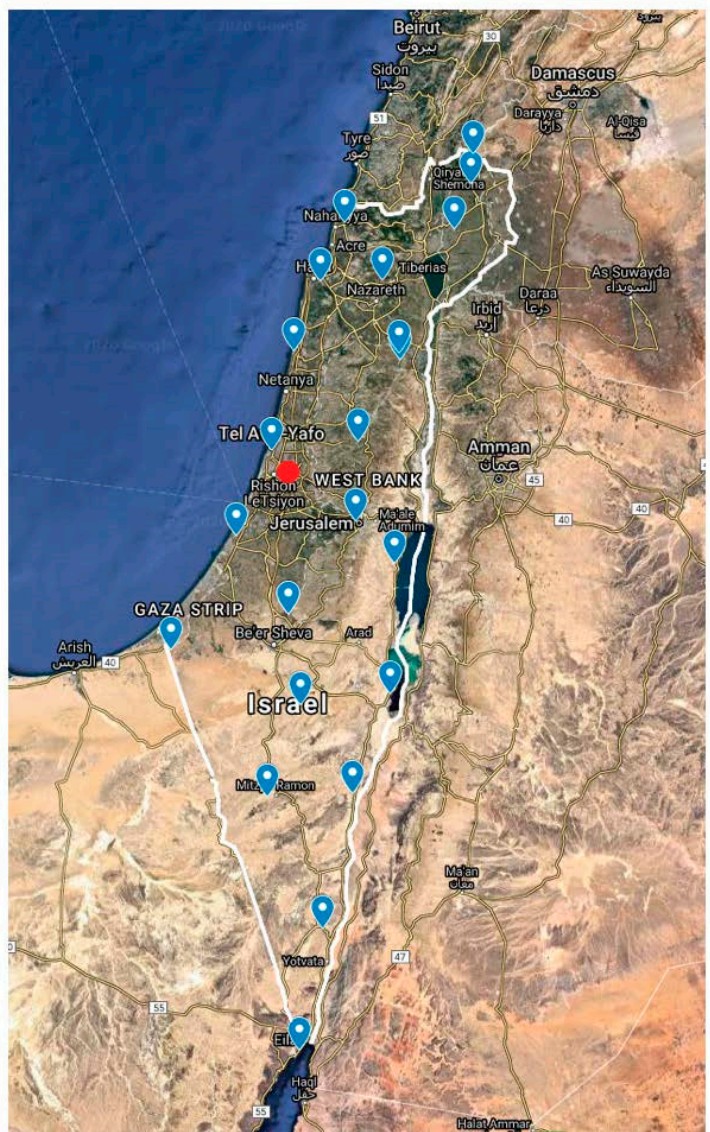

**Figure 1.** Israel's survey of Israel–active permanent network (SOI-APN) global positioning systems (GPS) network. The Survey of Israel (MAPI) is maintaining the network, which consists of 24 permanent geodetic GPS receivers. The Bed Dagan radiosonde location is marked by the red circle.

The outline of the paper is as follows. Section 2 presents the data and techniques used for producing the high-resolution 2D IWV distribution maps derived from GPS tropospheric path delays, followed by the augmented high-resolution 2D GPS-IWV using spatio-temporal cloud distribution extracted from METEOSAT satellite data. Section 3 describes the WRF model setup and data assimilation technique. Section 4 presents WRF forecasts and their verification versus radiosondes and 2D IWV maps showing the impact of assimilation. The discussion and conclusions follow in Section 5.

## 2. Measurements and Techniques

### 2.1. 2D IWV Distribution Maps Derived from GPS Tropospheric Path Delays

For the present study, we used the data obtained from the Israeli GPS network (SOI-APN) that includes more than 20 stations, covering the entire Israeli area [31,32]. These geodetic GPS stations record raw data in RINEX format. The GPS measurements

used in the current study were analyzed using the NASA-JPL GIPSY-X precise point positioning (PPP) platform. Thus, using the GIPSY-X PPP software and products, it is possible to calculate the zenith total delay (ZTD) (which is composed both from the zenith wet (ZWD) and hydrostatic (ZHD) delays). We used a 7° minimum elevation cut-off for the satellite observations alongside the Vienna Mapping Function 1 (VMF1) grid [33]. Zenith hydrostatic delay (ZHD) estimations were applied every 6 h using the VMF1 Grid. To enable time-varying behavior, GIPSY-X software handles the lower atmospheric zenith path delay and gradients as stochastic parameters, which are treated during each time step as constant, can be modified from a one-time step to another. Once the GIPSY-X software handles a measurement (while updating the estimated parameters), a time update is performed, introducing process noise to the parameter uncertainties, thus taking into account all mismodeled effects [34–36]. By [3], once the zenith dry delay (also known as the ZHD) is accounted for using surface temperature and pressure data, the amount of water vapor is directly proportional to the value of the ZWD:

$$\frac{PW}{ZWD} = \kappa/\rho \tag{1}$$

where

$$1/\kappa = 10^{-6} \left( k_3 T_m + k'_2 \right) R_v \tag{2}$$

$R_v$ is the specific gas constant, $k_3$ and $k'_2$ are some constants, $T_m$ is the weighted atmospheric mean temperature and $\rho$ is the water density. The ZWD values were calculated for the wet winter season and relatively dry spring season. For a more comprehensive description of this procedure, along with the precise values for the tropospheric parameters, see Leontiev and Reuveni [31,32].

After the ZWD parameters are estimated from the entire SOI-APN GPS geodetic network, a 2D IWV distribution grid can be extracted while applying different interpolation techniques. In the current study, we applied the Kriging interpolation (rather than the Delaunay interpolation) with the GPS-IWV estimations, which yield the most precise results compare with in-situ radiosonde measurements [32]. The interpolation was performed within the area of 29–35°N and 34–36°E, covering the entire Israel area along with some parts of Lebanon, Syria, Jordan, and Egypt (Figure 2a). Furthermore, the GPS derived ZWD or IWV are interpolated between the stations with taking into account the elevation values, which can be taken from any Digital Elevation Model (DEM), for example, SRTM1 or SRTM3. Each map consists of 200 × 600 points, which correspond to an area of 2° × 6°, longitude by latitude, respectively. These sets of data were assimilated into the model. Furthermore, the simplest way to determine whether the regional 2D IWV distribution map (constructed from triangulating all available GPS data) are sufficiently accurate, is by comparing the IWV values above the same location where the radiosonde observations are taken (i.e., at Bet Dagan site, which is located at the center of the country with latitude 32.00° and longitude 34.81°). For a more detailed description of the methodology, see Leontiev and Reuveni [32].

*2.2. Augmented 2D IWV Distribution Maps Using Gps-Iwv Estimations and Spatio-Temporal Cloud Distribution Extracted from Meteosat Satellite Data*

Using the methodology described in Leontiev and Reuveni [31,32] it is possible to obtain further enhancement of the 2D IWV distribution maps using METEOSAT satellite data. The above-mentioned strategy is constructed first by calibrating METEOSAT 7.3 μm WV observed brightness temperature values using the mathematical dependency between the IWV absolute values extracted from GPS ZWD and the METEOSAT-11 data (Figure 2b). The red color in Figure 2b represents the estimated GPS IWV values taken from a GPS station at clear sky conditions but with a line of sight (from the satellite to the station), which intersects the clouds (i.e., where cloud located between satellite and GPS station, thus the temperatures obtained from the METEOSAT-11 images are more than 5 K cooler relative to the Israel Meteorological Service (IMS) ground measurements and can be removed).

The green color represents estimated GPS IWV values taken from stations in cloudy conditions but with a line of sight, which does not intersect the clouds (i.e., the cloud located above the station, but not between the satellite and station, thus the estimated METEOSAT-11 temperatures are approximately equal to the IMS temperatures but the estimated IWVs are lower than the actual values and are also can be removed). The blue color represents clear sky conditions from which the mathematical dependency was extracted. We then used the surface temperature differences between the Israel Meteorological Service (IMS) ground station measurements and METEOSAT-11 10.8 μm infra-red (IR) channel to identify spatio-temporal cloud distribution structures (Figure 2c). For the last step, the identified cloud features were mapped into the GPS-IWV distribution map while performing the interpolation between the adjusted GPS station inside the network (Figure 2d).

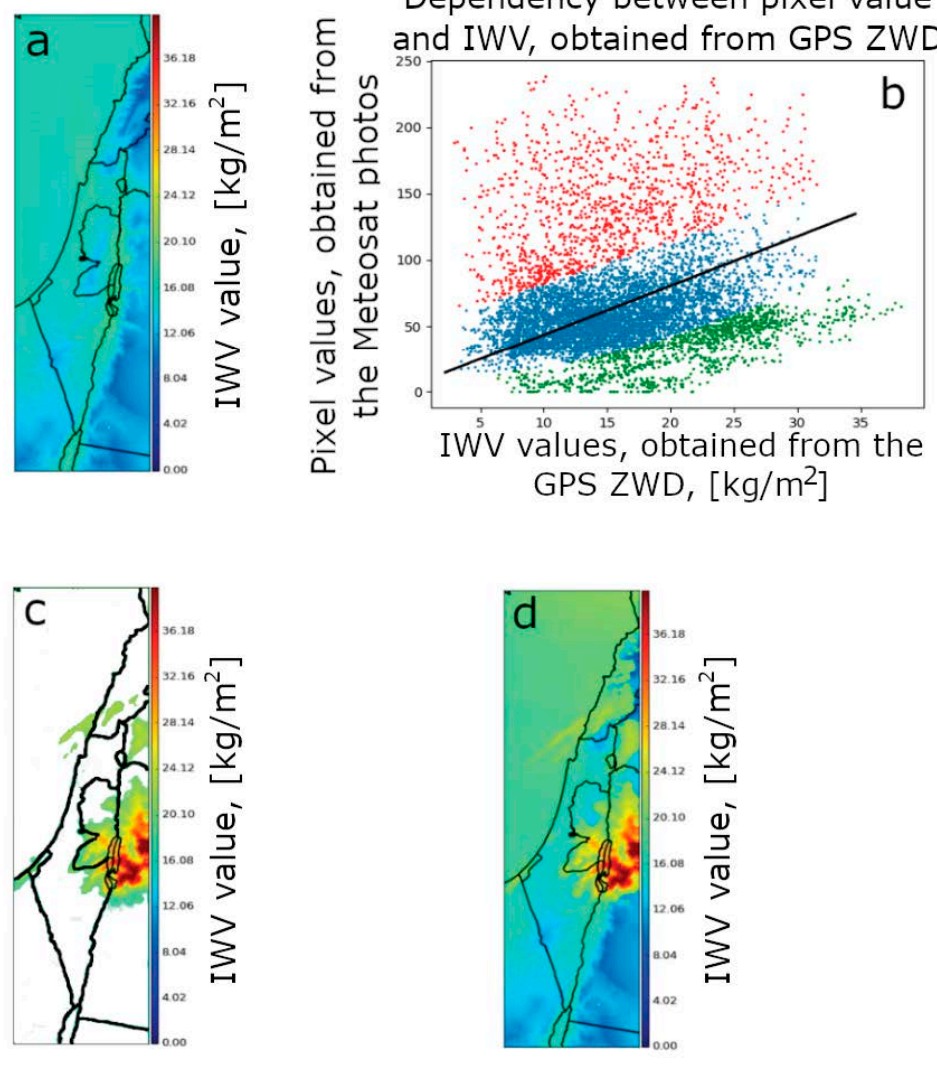

**Figure 2.** (**a**) GPS-IWV distribution map using Kriging interpolation. (**b**) Extracting the mathematical dependency between METEOSAT 7.3 μm water vapor (WV) pixel value and integrated water vapor (IWV), obtained from GPS ZWD. The different colors for points correspond to 3 main regions, which depend on the station location, weather conditions (mainly cloudy), and satellite position. (**c**) Using the surface temperature differences between the Israel Meteorological Service (IMS) ground station measurements and METEOSAT-11 10.8 μm infra-red (IR) channel to identify spatio-temporal cloud distribution structures [32]. (**d**) The identified cloud features are mapped into the GPS-IWV distribution map when performing the interpolation between adjusted GPS station inside the network.

### 3. WRF Model Setup and Data Assimilation

*3.1. WRF Setup*

The WRF model is considered a powerful tool for investigating weather and processes in the troposphere [37,38]. The WRF model solves the compressible non-hydrostatic atmospheric equations in flux form on a mass-based, terrain-following, vertical coordinate system. High-resolution global datasets are used to define the model topography and other static surface fields. For a complete description of the WRF modeling system, see, e.g., [6]. The WRF model has a nesting grid capability that allows zooming into a sub-region with a high horizontal resolution by generating a series of higher resolution nested grids within the coarser parent grids. Besides, WRF includes a complete suite of physics schemes that account for the important atmospheric and land–surface physical processes.

A single model domain with a grid size of 11 km (Figure 3) was chosen. The studied period extended from 1 January 2018 to 31 May 2018, i.e., during the winter–spring season when the frequency of high cloud coverage and precipitation is the highest compared to other seasons.

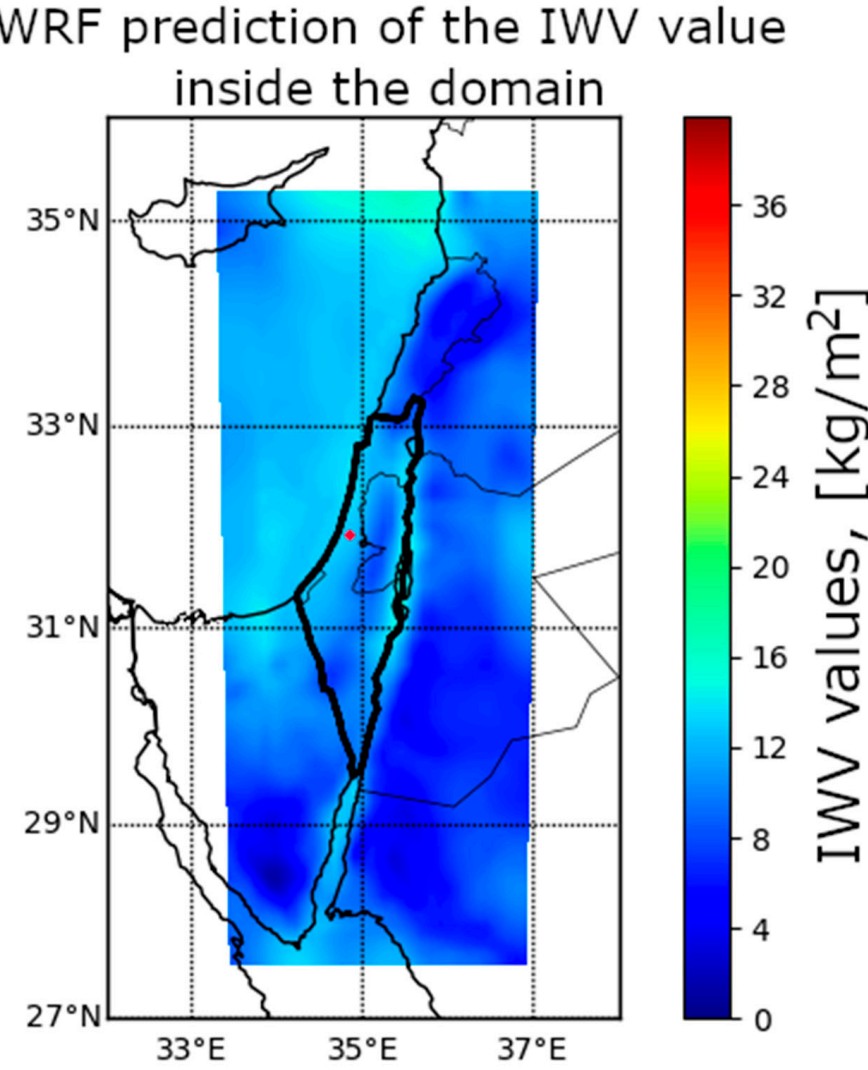

**Figure 3.** Calculated IWV distribution inside the Weather Research and Forecast (WRF) model domain is represented by the colored area. The grid size was 33 over 80 points, chosen grid size was 11 km, where the reference longitude and latitude of Lambert projection were equal to 35.178 and 31.428 degrees, respectively. The location of the Israel Meteorological Service radiosonde at Bet Dagan is indicated by the red circle.

The configuration of the WRF (version 4.0) model physical parametrization, used in our research, included the surface layer fifth-generation Pennsylvania State University–NCAR Mesoscale Model (MM5) scheme [39,40], which is the best option for hydrometeors investigation. For the short-wave radiation Dudhia [41] and long-wave, we used the Rapid Radiative Transfer Model (RRTM) [42]. For the planetary boundary layer (PBL) parameterization, the non-local Yonsei University PBL scheme was used [43]. Furthermore, we used 35 vertical levels up to 25 hPa with 20 of them within the lowest 1.5 km above the surface, with the GFS model analysis and forecasts [44] (its description can be found at https://www.ncdc.noaa.gov/data-access/model-data/model-datasets/global-forcast-system-gfs) for the atmosphere and soil initial and lateral boundary conditions.

### 3.2. WRF Data Assimilation Technique and Implementation

The WRF Data Assimilation (WRFDA) is a supplement module to the WRF model which allows 3DVAR and 4DVAR data assimilation of conventional and remote sensing observations (e.g., pressure, radio occultation data, IWV/ZWD measurements, etc.) Here we used a 3DVAR assimilation technique, the cost function of which is presented below:

$$J(x) = (x - x_b)^T B^{-1}(x - x_b) + (y - H(x))^T R^{-1}(y - H(x)), \qquad (3)$$

where $B$ denotes the background error covariance, $R$ the observational error covariance, $x$ and $x_b$ are the model state vector and background vector respectively, $y$—the observation vector, defined at the time of data assimilation, $H$ is the linear operator, which transforms parameter from the model space to the observation space. In order to calculate background errors matrix, the analysis-ensemble method was used [45] and the $R$ matrix was taken to be diagonal. In this study the assimilated data was limited only by the GPS derived IWV, in contrast to many other studies, for example [46,47], where frequent and dense radiosonde data was also assimilated. The Israel Meteorological Service (IMS) radiosonde data in Israel is available two times per day, however, there is only one radiosonde location (see Figure 3). Therefore, the radiosonde data was used for the verification of the model forecasts only. Surface observations are scarce as well, as opposed to Europe and the USA. Moreover, surface observations are not expected to contribute significantly when assimilated into a model run at 11 km grid-size due to the local representativeness of such observations, in particular at areas characterized by complex terrain and land use heterogeneity as is the studied area, see e.g., [48]. Since the IMS radiosondes provide measurements every 12 h, at 00:00 and 12:00 UTC, WRF model initialization times without assimilation were chosen to provide 3, 6, and 12 h lead-time forecast at radiosondes times. The data assimilation runs were similarly initialized every 3, 6, and 12 h before radiosondes times and IWV 2D maps were assimilated at initialization times only.

Three types of WRF runs were done. The first type consists of control runs of the WRF model without any assimilation (defined here as the Control runs). The second type of runs included GPS-IWV data assimilation only (defined here as AssimGPS runs). In contrast to other similar studies [49,50], where only single GPS derived IWV or ZWD/ZTD values were assimilated, in this study, we assimilate the quasi-continuous map of IWV consisted of 120,000 points, as described in Section 2.1. Lastly, the third type of runs assimilated into WRF the GPS-IWV derived maps augmented by the METEOSAT data, as described in Section 2.2 (defined here as AssimGPS-METEOSAT). The free forecasts for 3, 6, and 12 h lead times for the three types of runs were compared to the radiosonde data using the model-grid point closest to the radiosonde location. We calculated root mean square errors (RMSE) with confidence intervals, mean error (ME), and linear correlation coefficient (C) with respect to the radiosonde's observations for each WRF free-forecast lead times. Confidence intervals at 95% significance were calculated by using the bootstrapping method with 10,000 samples. These parameters are presented in Table 1 and will be discussed in Section 4.

**Table 1.** Comparison of RMSE, ME and C values for WRF Control, Assim-GPS, and AssimGPS-METEOSAT runs.

|  | **3 h** | **6 h** | **12 h** |
|---|---|---|---|
| **1. Control** |  |  |  |
| **RMSE, mm** | 2.534, (0.439, 0.437) | 2.577, (0.429, 0.434) | 2.639, (0.332, 0.323) |
| **ME, mm** | 1.69 | 1.828 | 1.317 |
| **C** | 0.87 | 0.88 | 0.8 |
| **2. AssimGPS** |  |  |  |
| **RMSE, mm** | 2.246, (0.291, 0.301) | 2.381, (0.299, 0.291) | 2.478, (0.266, 0.328) |
| **ME, mm** | 1.473 | 1.256 | 1.383 |
| **C** | 0.89 | 0.84 | 0.84 |
| **Improvement of RMSE with respect to the Control run (%)** | 11 | 7.5 | 6 |
| **3. AssimGPS-METEOSAT** |  |  |  |
| **RMSE. mm** | 1.754, (0.235, 0.195) | 1.778, (0.210, 0.207) | 1.819, (0.262, 0.202) |
| **ME, mm** | 0.653 | 0.806 | 0.569 |
| **C** | 0.89 | 0.9 | 0.89 |
| **Improvement of RMSE with respect to the control run (%)** | 30 | 31 | 31 |

Furthermore, these types of runs for WRF analysis and first guess are represented in Figure 4a,b. Figure 4c represents the comparison of interpolated GPS 2D maps with the radiosonde data. As can be noticed, WRF analysis runs are very mamillar to the GPS-IWV 2D maps values, however still showing higher bias and error values. These results are summarized in Table 2.

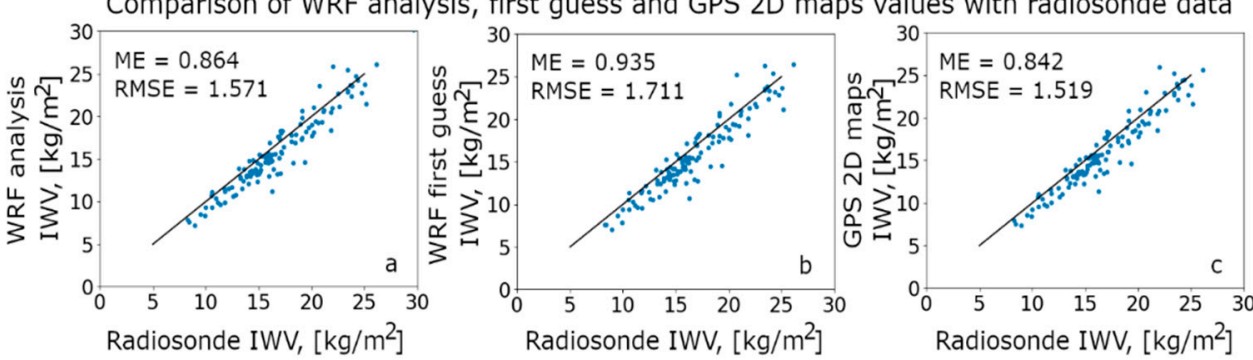

**Figure 4.** (**a**) Comparison of WRF model analysis calculation vs. radiosonde data. (**b**) Comparison of WRF model first guess vs. radiosonde data. (**c**) Estimated 2D GPS-IWV maps vs. the radiosonde data.

**Table 2.** ME and root mean square error (RMSE) values for first guess, analysis, and GPS 2D maps in comparison to radiosonde values.

|  | **ME, mm** | **RMSE, mm** |
|---|---|---|
| **WRF first guess** | 0.935 | 1.711 |
| **WRF analysis** | 0.864 | 1.571 |
| **GPS IWV 2D maps** | 0.842 | 1.519 |

## 4. Results and Verification

### 4.1. Verification of Control WRF Forecasts

Using the above-mentioned strategy, we first run the WRF model in "control" mode, i.e., without assimilation of observations to produce forecasts of the IWV field covering the entire area of Israel and neighboring areas. The domain configuration with an example of the IWV WRF forecast is depicted in Figure 3. Figure 5a–c present scatter plots of IWV from the

free forecasts vs. radiosonde observations for 3, 6, and 12 h lead times, respectively. Section 1 of Table 1 summarizes ME, RMSE, and C values as a function of lead time for the control forecasts. As expected, errors increase with lead time, following previous studies [51–53].

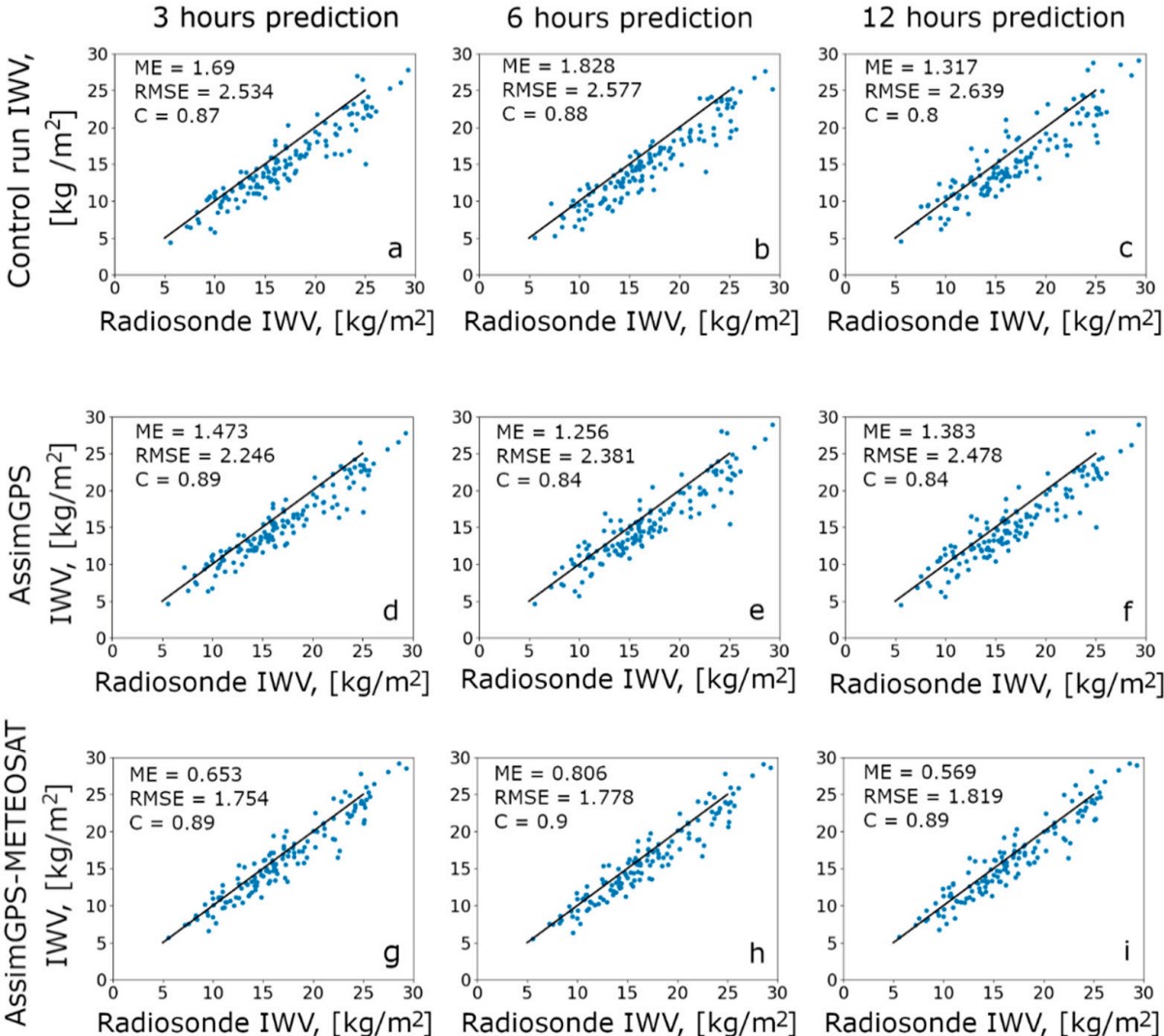

**Figure 5.** (**a**–**c**) Comparison of Control runs with radiosonde measurements for 3, 6, and 12 h, respectively. (**d**–**f**) Comparison of AssimGPS with radiosonde measurements for 3, 6, and 12 h, respectively. RMSE is reduced by ~6–11%. (**g**–**i**) Comparison of AssimGPS-METEOSAT with radiosonde measurements for 3, 6, and 12 h, respectively. RMSE is reduced by ~30%.

### 4.2. Verification of AssimGPS Forecasts

Figure 5d–f present scatter plots of 3, 6, and 12 h free forecasts of IWV from the AssimGPS runs vs. radiosondes observations, respectively. RMSE, ME, and C values for the AssimGPS runs vs. radiosonde observations are summarized in Section 2 of Table 1. As can be noted, RMSE for the AssimGPS runs is decreased up to 11% (for the 3 h forecasts) with respect to that of the Control runs.

### 4.3. Verification of AssimGPS-METEOSAT Forecasts

For the next stage, we run the WRF model with the assimilation of the augmented 2D GPS-IWV regional maps using spatio-temporal cloud distribution extracted from METEOSAT-11 (i.e., AssimGPS-METEOSAT runs) to produce 3, 6, and 12 h forecast, and compared them with the IMS radiosonde measurements. We note that Control, AssimGPS,

and AssimGPS-METEOSAT runs were based on the same timeframe. Figure 5j–i present scatter plots of 3, 6, and 12 h free forecasts of IWV from the AssimGPS-METEOSAT runs vs. radiosonde observations with summarized values of RMSE, ME, and C in Section 3 of Table 1. As can be noted, RMSE between the AssimGPS runs is reduced by ~ 30% for that of the Control runs. A summary of all the results is presented in Figure 6. The ME and RMSE values, which are represented in Table 1, show that both ME and RMSE values for the AssimGPS runs are slightly decreased with respect to the Control runs, while for the AssimGPS_METEOSAT runs they are significantly decreased.

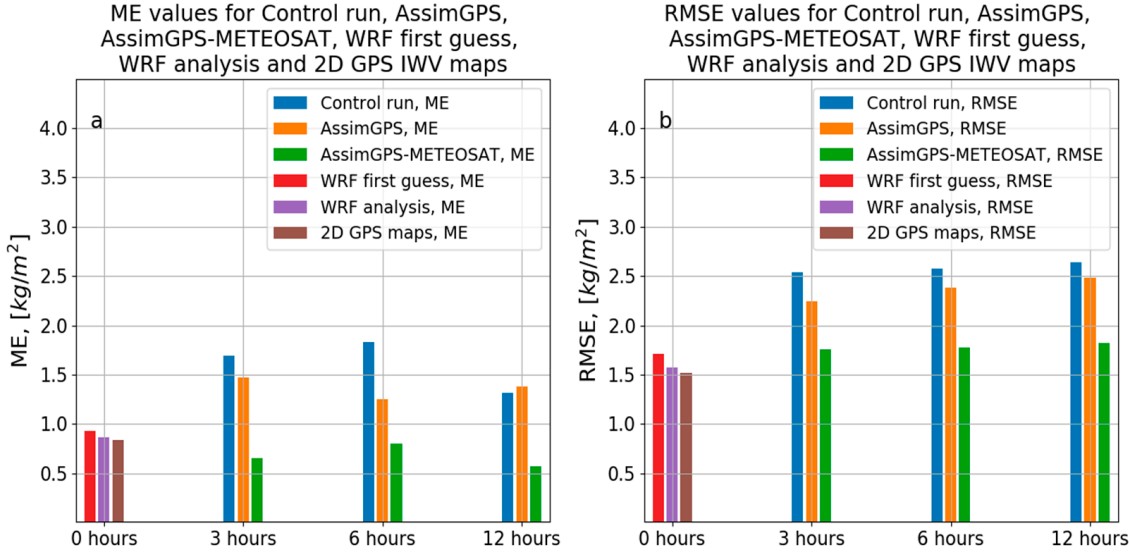

**Figure 6.** (**a**) Comparison between ME values for Control run, AssimGPS, WRF first guess, WRF analysis, 2D GPS IWV maps and AssimGPS-METEOSAT, with respect to the radiosonde data. (**b**) Comparison between RMSE values for Control run, AssimGPS and AssimGPS-METEOSAT, with respect to the radiosonde data.

### 4.4. Verification of AssimGPS and AssimGPS-METEOSAT by Using 2D GPS IWV Maps

Since the Israeli radiosonde network is represented by a single station, it is impossible to estimate the enhancement of WRF runs over the entire domain. The station is located near the coastline (a distance of about 15 km, see Figure 3), several meters above sea level. There are no obstacles, which can hold wet masses of air, so it is impossible to claim that WRF results are improved over the entire domain. Israeli area includes coastline, inland, mountains, and rifts, where wet air can be accumulated due to topography. However, it is possible to use the 2D GPS IWV maps for verification purposes. As opposed to the radiosondes, 2D GPS IWV maps were used to verify WRF forecasts only, not WRF analysis. The maps used for verification are valid at verification time, not at assimilation time. These are different from the assimilated maps. The verification procedure is the same as for the radiosonde one. GPS-METEOSAT IWV values, estimated at numerous different points on the map, are compared with the WRF runs values at the same points; this procedure is performed for every GPS-METEOSAT estimated map and the corresponding WRF run. As a next step of the verification procedure, it is possible to obtain scatter plots similar to the scatter plots presented in Figure 5, each plot corresponding to a specific point in the WRF domain. The RMSE and ME values for each point can be depicted on the map. This leads to a better understanding of the spatial improvements of the suggested methodology. The results are shown in Figures 7 and 8. As can be noticed, a significant spatial enhancement is achieved (in terms of ME and RMSE values) for the AssimGPS-METEOSAT runs as compared to Control and AssimGPS runs.

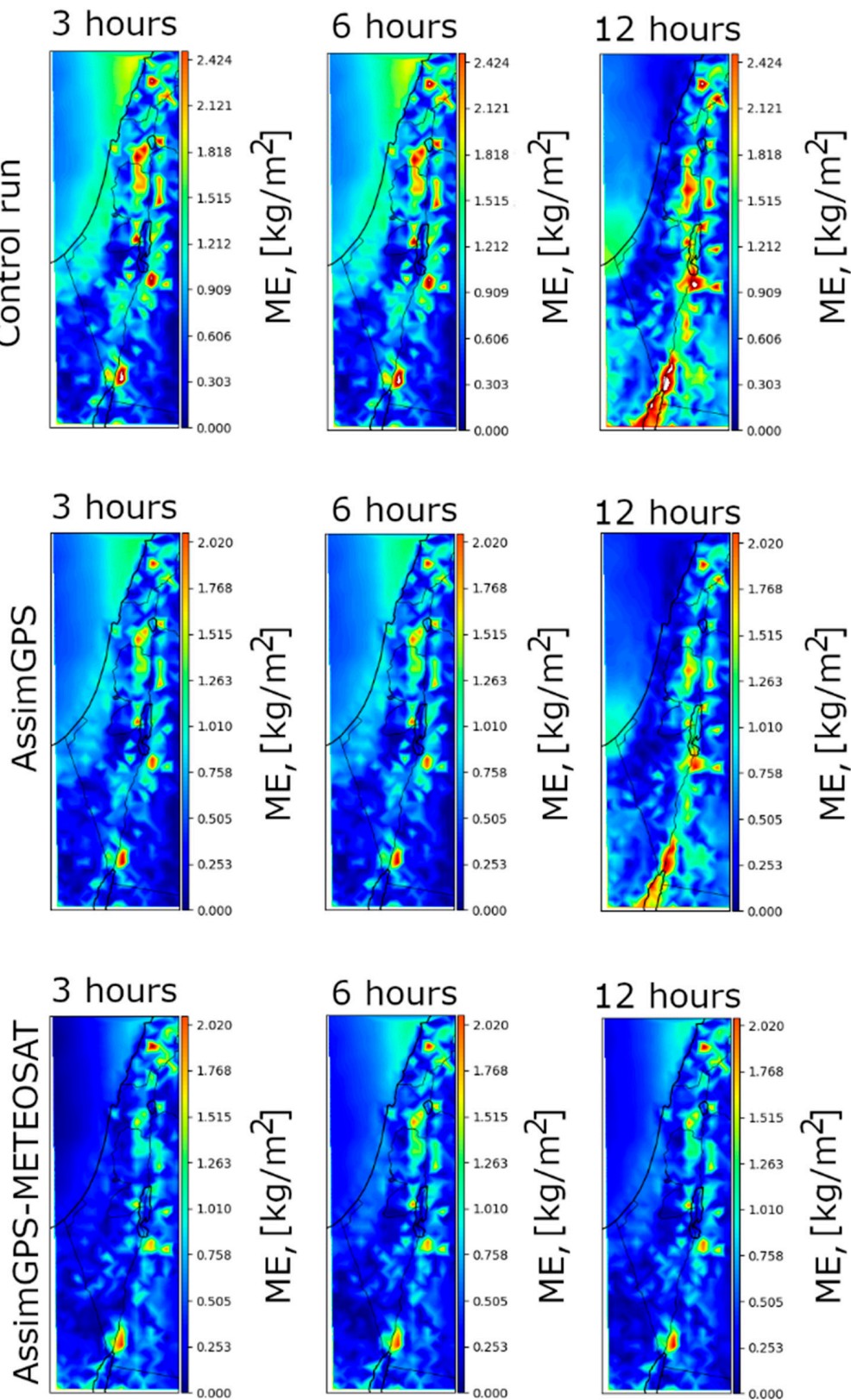

**Figure 7.** Mean error (ME) maps between the 2D GPS-IWV-METEOSAT and different types of WRF forecasts.

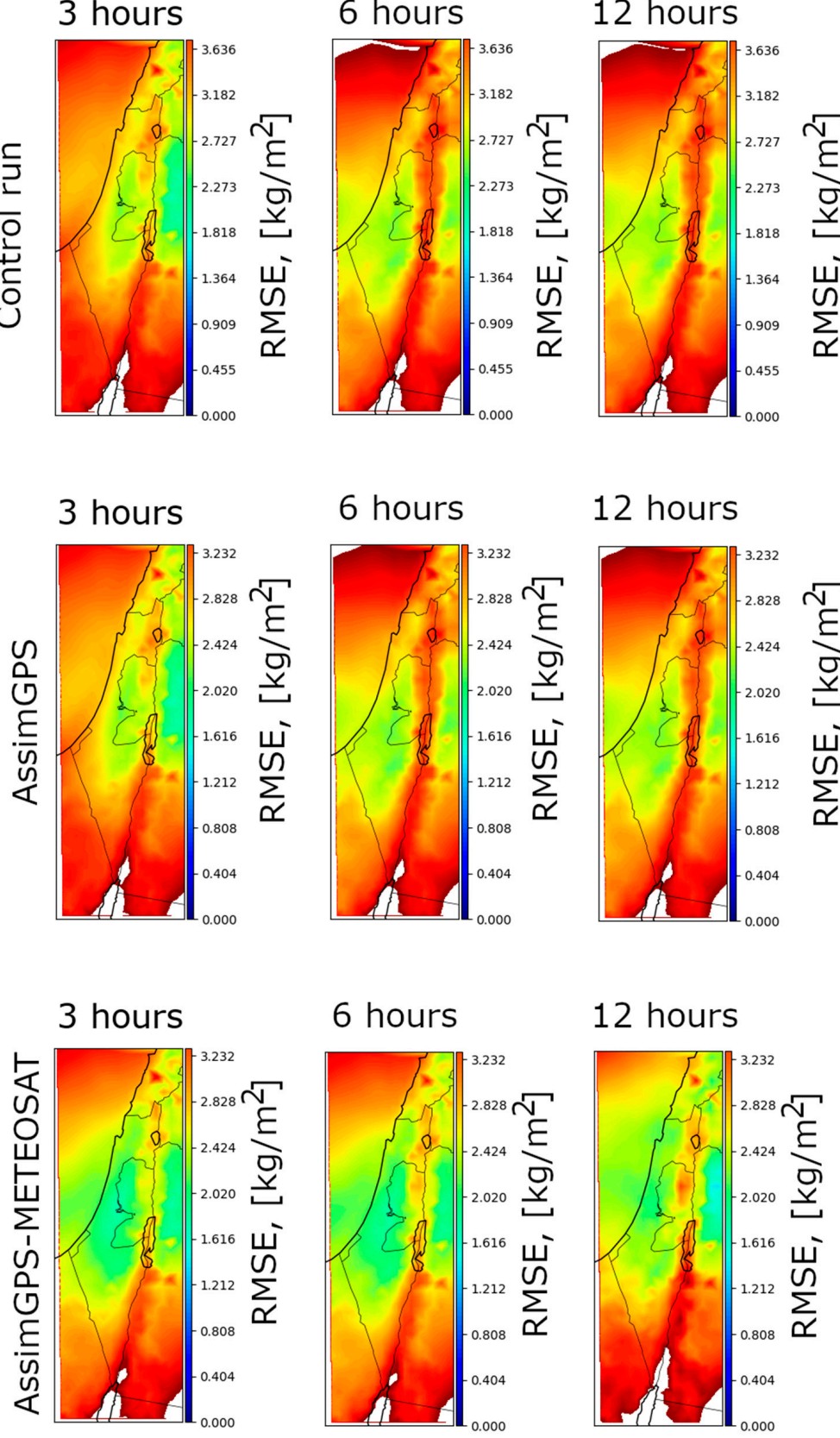

**Figure 8.** Root mean square error (RMSE) maps between the 2D GPS-IWV-METEOSAT and different types of WRF forecasts.

### 4.5. Verification of AssimGPS and AssimGPS-Meteosat versus Vertical Profile of Relative Humidity

The radiosondes at Bet-Dagan were used to verify the simulated profiles of relative humidity (RH). Figure 9 presents these results. The technique was the same as the technique presented in Figure 7, and described in details above. It is seen from Figure 9, that the RMSE decreases for AssimGPS and AssimGPS-METEOSAT runs at the lower part of the troposphere, where values of RH are considerable. In particular, we observe significant improvement at 12 h lead time, as the control run suffers from large RMSE values near the surface (up to 60%). As in the verification results shown in Sections 4.1–4.4, AssimGPS-METEOSAT runs show the most significant improvement with respect to the control runs, cutting errors by up to half of their values.

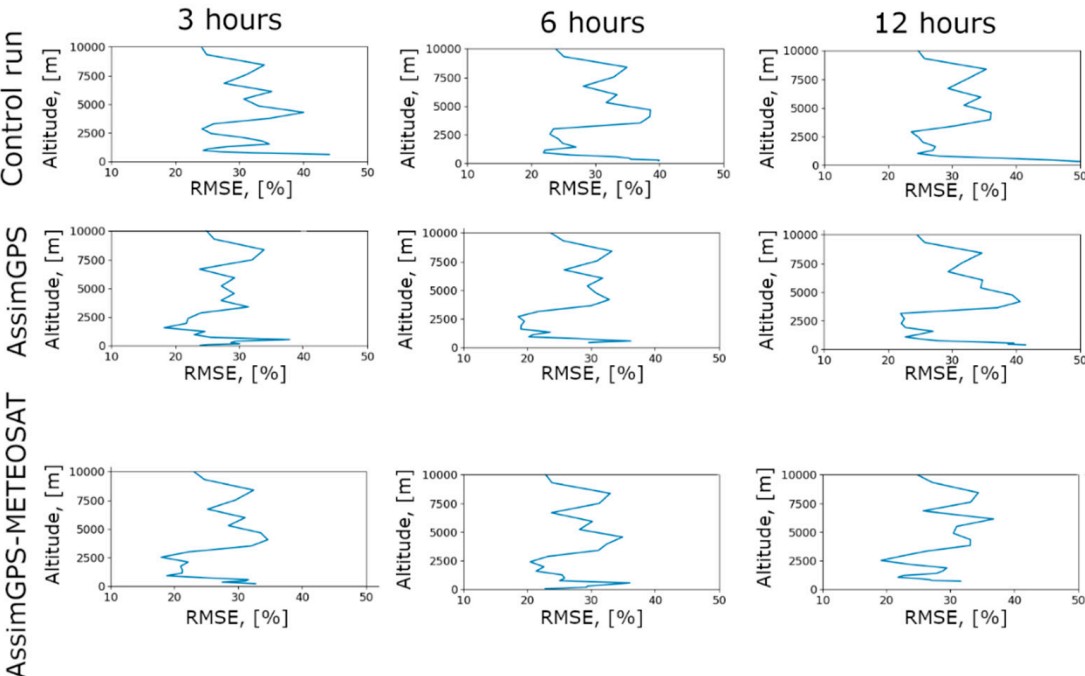

**Figure 9.** Vertical profile of RMSE of RH for the different types of runs and lead times.

## 5. Discussion and Conclusions

Here, we present a new methodology for improving the WRF model forecasts skill using a data assimilation procedure, which combines estimated 2D IWV regional maps derived from both GPS tropospheric path delays and METEOSAT-11 data. Previous studies, which used point measurements data assimilation from radiosondes measurements or GPS zenith wet delay (ZTD) estimations, demonstrated small improvements of WRF forecasts. Using the suggested technique, our results indicate a decrease of up to 30% in RMSE (when verified versus radiosonde data) for forecasts from AssimGPS and AssimGPS-METEOSAT runs relative to the Control runs without assimilation. The calculated RMSE and ME indicate that assimilating GPS 2D maps, in combination with METEOSAT-11 data, significantly reduces the RMSE values, and to a lesser extent ME values. A summary of all the results in comparison with the radiosonde data is presented in Figure 6.

Figures 7 and 8 present the spatial distribution of ME and RMSE inside the WRF domain. The RMSE is significantly reduced for the AssimGPS-METEOSAT maps along the coastline, Jordan Rift Valley, and Dead Sea areas, as well as the Golan Heights. ME values are significantly reduced near the Jordan Rift Valley, surrounded by mountains from the East and West. In general, significant improvements are taking place near topographic landform areas (i.e., mountains and valleys). Our results demonstrate that these improvements are caused by the assimilation of high resolution observational data (up

to 100 m grid, which is directly measured near GPS stations, located at the Jordan Rift and the Golan height) as opposed to the use of the coarse GFS model data only (0.25°, about 25–30 km, grid). The grid of GFS model data covers up to 8 points over Israel (in the East-West direction), whilst our high-resolution grid of GPS IWV maps covers more than 1000 points.

Furthermore, these improvements may be associated with the frontal Mediterranean Sea Breeze (MSB) due to the fact, that the highest improvements are along the coastline, Jordan Rift Valley and the Dead Sea. The MSB propagates from the coastline eastward inland combined with northern winds enhancement due to Coriolis force, which brings moisture inland as recently suggested by Ziskin et al. [54], and observed (using in-situ and ground remote-sensing) and simulated using WRF model by Kunin et al. [55]. The initialization times cover the period between 03:00–15:00 LT. This period includes times during which the MSB propagates from the coastline to highland stations. Detailed information about the moisture brought by the MSB is not available in the coarse GFS analyses, demonstrating again the importance of the assimilation of high resolution IWV maps into WRF at initialization times. Further research work may include assimilating 2D GPS-METETOSAT maps of IWV into WRF under different meteorological scenarios, e.g., for different seasons, sky, and weather conditions, while considering all the necessary adjustments for achieving the best performance for enhancing the suggested technique (e.g., assimilation cycling). In addition, given that AssimGPS-METEOSAT runs reduces the RMSE, one possible direction for future work may be to focus on studying the sources of these errors in order to reduce them furthermore, for example, by using maps with higher spatial resolution.

**Author Contributions:** All authors have made significant contributions to the manuscript. Conceptualization, Y.R.; Formal analysis, A.L.; Methodology, Y.R.; Validation, A.L., D.R.-E. and Y.R.; Visualization, A.L.; Writing—original draft, A.L. and Y.R.; Writing—review & editing, D.R.-E. and Y.R. All authors have read and agreed to the published version of the manuscript.

**Funding:** This work was funded by the Israeli Ministry of Science, Technology & Space grant 3-14814.

**Institutional Review Board Statement:** Not applicable.

**Informed Consent Statement:** Not applicable.

**Data Availability Statement:** Continuous GPS data were provided by SCIGN, operated by the Scripps Orbit and Permanent Array Center (SOPAC).

**Conflicts of Interest:** The authors declare no conflict of interest.

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
