# Peer review of "On the Potential of Improving WRF Model Forecasts by Assimilation of High-Resolution GPS-Derived Water-Vapor Maps Augmented with METEOSAT-11 Data"

_remotesensing, doi:10.3390/rs13010096_

Round 1
Reviewer 1 Report
It is a good idea to assimilate the 2D PWV map enhanced by the METEOSAT in WRF. However, as for the experiment and verifications in this paper, there are several major problems that require addressed or clarified.
- L168: how did you map the cloud feature into the GPS-IWV distribution map? How did you know the correctness after the augmentation of METEOSAT cloud features? Please provide more details because this is very critical in your study.
- I would like to suggest you to assimilate other observations except IWV in your control run, e.g., the SYNOP, METAR, SHIP, BUOY, etc., which would be more realistic and convincing.
- 4.4.: the 2D GPS IWV maps are not independent, and cannot be used as reference to verify the WRF results. Maybe you can pick part of GPS stations as independent stations only for verification instead of assimilation.
- Only the IWV were verified in the study. It would be better if the authors can step forward, to check the precipitations in some representative cases.
Other minor problems includes,
- L58: ‘Yan et . El.’ To ‘Yan et al.’
- L88: provide references for this number (5-10%)
- The title of 2.1.: ‘D IWV’ to ‘2D IWV’
Author Response
We would like to thank the reviewer for the time and effort dedicated to reviewing our paper. All comments, suggestions, and questions were carefully considered, and all necessary corrections were made in the revised manuscript.

Reviewer 2 Report
Review of Leontiev et al. for Remote Sensing
The authors describe a very interesting technique for improving the analysis and short-term forecasts of humidity by assimilating GPS measurements of IWV combined with satellite imagery. I recommend that this article be reconsidered after major revision.
- The English needs work, mainly in terms of grammar and usage. The science is laid out logically, but the wording is often awkward.
- Line 104: Should this be “METEOSAT-11” rather than “METEOSAT-10”?
- Section 4.3: Radiosonde data are used for verification without providing any description of the data. Which radiosonde type is used by IMS? Radiosonde humidity sensors vary widely in their characteristics, with some types having significant biases that might affect IWV. In addition, the vertical resolution of the radiosonde data might also play a role in how accurately IWV can be estimated. There is a large difference between BUFR radiosonde messages with hundreds to thousands of levels and the standard TEMP messages that typically provide fewer than 100 levels with humidity observations. The precision of the humidity values also differs between these two formats, with BUFR values having more precision. In other words, how confident are the authors in the radiosonde estimates of IWV? How much uncertainty is present?
- Figures 7 and 8: the color scale in these figures is different for the control run than for the two types of AssimGPS runs, which makes comparison more difficult. I recommend that the authors revise these two figures to use the same color scale for the control run as was used for the other panels within that figure (or equivalently, change the color scale for the AssimGPS panels to agree with the control run). I realize that Figure 7 uses a different color scale than Figure 8 and don’t have a problem with that.
- As I understand the procedures, the authors are allowing the 3DVAR to distribute differences in IWV vertically in terms of the analysis control variable for humidity. How did the AssimGPS runs change the vertical distribution of humidity compared to the control run? Since these experiments led to differences at the radiosonde site, it would be useful to include a figure that shows the vertical distribution of mean and RMS differences with respect to the radiosonde data in terms of mixing ratio or specific humidity.
Author Response

(The authors gave the same response as above.)

Reviewer 3 Report
This paper describes the impact of assimilating GNSS IWV, modified by Meteosat IWV data, in the performance of WRF forecasts of IWV. The synergy between GNSS and Meteosat appears the main novelty of the study, but this idea has been published by the authors in two referenced papers so the novelty here is only in the application. The paper is well written but not very well explained in some essential details.
I could not really understand the way Meteosat data modifies the IWV fields from GNSS. I know it is described in the previous papers, but a simple qualitative and concise paragraph would help. Most of the description if oddly provided in very long figure caption (Fig. 2) but I found it to be a little confusing. Does it say that Meteosat data is used to locate the cloud boundaries and then specific humidity is there adjusted to saturation and IWV computed? Or is it something else?
I also have a couple of concerns related with the statistics. It appears that IWV data from the GNSS network is used both as data to assimilate and as the ground truth to compute the error statistics. If that is not the case, please clarify (cf lines 305ff). If it is, we have a problem, because it would make the error statistics unacceptable. The second point concerns the high resolution grid of IWV data: it appears to be just the result of interpolation with a DTM model. Why do you need such resolution for WRF at 11km? Or is WRF at 1km? The horizontal interpolation would appear to introduce a lot of redundancy. Is this what is called the quasi-continuous map of IWV? why would it be better than a coarser grid given the inherent smoothness of GNSS data?
Figure 4. What is the difference between WRF analysis and WRF first guess?
What is Mean Error? Is it mean absolute error or mean bias?
Without understanding the ground truth for the computation of the errors I can not understand the relevance of the results...
I could suggest that precipitation could be an independent, but more challenging, way to assess your results. I suggest you to look at INSAR data assimilation.
Author Response

(The authors gave the same response as above.)

Reviewer 4 Report
The authors present a method for improving the accuracy of weather model forecasts with measurements of the water vapor with GPS and also data from METEOSAT. Their approach is clever and produces probably better results than the model forecasts alone. But this is not certain, because the authors do not provide any estimate of uncertainties. They calculate errors that are indeed smaller with AssimGPS-METEOSAT than with AssimGPS and much smaller than with the control run, but the mean error is always smaller with 12 h than with 3 h, by up to 22%. So maybe if the authors had studied data from another year, they would have found that AssimGPS produced a smaller error than AssimGPS-METEOSAT. Therefore I believe that a careful calculation of the uncertainties for the values in Table 2 is needed, before this manuscript can be published.
Some minor comments:
"References" lacks some citations, e. g. Ehhalt et al. (2001) or Fisher (2001), and contains articles that are never cited, e. g. Huang et al. (2015). The order of references is sometimes not alphabetical, e. g. Lagasio et al. (2019) comes behind Lynn et al. (2005).
Line 50: "Bevis, 1992" is "Bevis et al., 1992".
Line 69, 87, 103, 128, 138, 165: Several abbreviations are explained many times, e. g. ZWD.
Line 99: Cresswell et al., 1999, not 1997
Line 206: 2014, not 2016
Line 247: Explain exactly what "mean error" is. What does it tell you that you cannot learn from RMSE? Is it always positive? Does it have an uncertainty?
Table 2: Improvement of RMSE w. r. t. the control run: 7.5%, improvement of ME for 6 h w. r. t. 3 h: 15% (AssimGPS). Why is this?
Line 291: (g) instead of (j)
Line 346: (2021) instead of (2020).
Author Response

(The authors gave the same response as above.)

Round 2
Reviewer 1 Report
All my comments have been properly addressed. I would like to suggest the acceptance in current form.
Reviewer 3 Report
The authors clarified a few issues, improving the paper. I still think it would need further validation to prove the robustness of the method.
Reviewer 4 Report
Not all of my minor comments were addressed, but the main problem I had, i. e. the absence of uncertainties, which meant than one could not judge whether the results were credible, is now solved. The remaining small issues concern discrepancies of the year of publications when cited in the text and in the list of references etc.